# Color Centers in Hexagonal Boron Nitride

**DOI:** 10.3390/nano13162344

**Published:** 2023-08-15

**Authors:** Suk Hyun Kim, Kyeong Ho Park, Young Gie Lee, Seong Jun Kang, Yongsup Park, Young Duck Kim

**Affiliations:** 1Department of Physics, Kyung Hee University, Seoul 02447, Republic of Korea; wraith01@khu.ac.kr (S.H.K.);; 2Department of Information Display, Kyung Hee University, Seoul 02447, Republic of Korea; 3Department of Advanced Materials Engineering for Information and Electronics, Kyung Hee University, Yongin 17101, Republic of Korea; junkang@khu.ac.kr

**Keywords:** hexagonal boron nitride, color center, light emission, quantum emitter

## Abstract

Atomically thin two-dimensional (2D) hexagonal boron nitride (hBN) has emerged as an essential material for the encapsulation layer in van der Waals heterostructures and efficient deep ultraviolet optoelectronics. This is primarily due to its remarkable physical properties and ultrawide bandgap (close to 6 eV, and even larger in some cases) properties. Color centers in hBN refer to intrinsic vacancies and extrinsic impurities within the 2D crystal lattice, which result in distinct optical properties in the ultraviolet (UV) to near-infrared (IR) range. Furthermore, each color center in hBN exhibits a unique emission spectrum and possesses various spin properties. These characteristics open up possibilities for the development of next-generation optoelectronics and quantum information applications, including room-temperature single-photon sources and quantum sensors. Here, we provide a comprehensive overview of the atomic configuration, optical and quantum properties, and different techniques employed for the formation of color centers in hBN. A deep understanding of color centers in hBN allows for advances in the development of next-generation UV optoelectronic applications, solid-state quantum technologies, and nanophotonics by harnessing the exceptional capabilities offered by hBN color centers.

## 1. Introduction

Since the first mechanical exfoliation of monolayer graphene in 2004 [1], atomically thin two-dimensional (2D) materials have shown their exotic physical properties, which are not present in bulk materials. Unlike the gapless semi-metallic properties of graphene, transition-metal dichalcogenides (TMDCs) [2], which are 2D semiconductors, have been found to have a direct bandgap, which corresponds to efficient optical emission from the visible to near-infrared range at the monolayer limit, with possible applications in advanced optoelectronic devices. Furthermore, the atomically thin insulating material hexagonal boron nitride (hBN) is composed of covalently bonded boron and nitride with an sp2 orbital, which is isomorphous with graphene [3]. The single-crystal structure of multilayer hBN is an AA’-type interlayer bonding structure, which appears as a hexagonal structure from the top view, as shown in Figure 1a,b, resulting in a large bandgap opening. Recently, hBN has served as an ideal encapsulation material for graphene in 2D van der Waals heterostructures and 2D semiconductor TMDCs due to its exceptional chemical and thermal stability against harsh environmental conditions, such as high temperatures. Furthermore, due to the absence of dangling bonds and the atomically clean interface, the single-crystal structure of hBN allows for limited intrinsic charge carrier scattering in 2D van der Waals heterostructure devices compared to polycrystal-structured materials, such as silicon dioxide [4,5,6].

The band structure of graphene, a hexagonal structure of carbon atoms, is characterized by the absence of a bandgap, leading to its semi-metallic behavior. On the other hand, the band structure of hBN includes a bandgap due to the presence of two different types of atoms, boron and nitrogen. As a result, hBN exhibits insulating behavior and a large bandgap opening. Figure 1c shows the band structure of hBN based on first-principles calculation, with the number of atomic stacked layers increasing from monolayer to bulk (infinite number of layers) [7]. Monolayer hBN exhibits a direct bandgap at the K point, with 6.47 eV bandgap energy. However, with the increase in the number of layers of hBN, the bandgap exhibits a transition in the conduction band, shifting to the M point while the valence band remains at the K point, resulting in the transformation from a direct bandgap to an indirect bandgap semiconductor.

To create high-quality hBN quantum devices, it is essential to use hBN with excellent material quality. Efforts to grow hBN have been attempted extensively, but challenges related to crystallinity issues and high impurity contents have often led to failures. In 2004, a group of scientists in NIMS (National Institute for Materials Science, Japan) led by T. Taniguchi [8] successfully achieved the large-scale growth of hBN using the HTHP (high-temperature high-pressure) method. The authors used the temperature gradient method under HP (4.0–5.5 GPa)/HT (1500–1700 °C) conditions using barium boron nitride (Ba3B2N4) as a solvent system to prepare samples of deoxidized hBN. They confirmed the high crystallinity and low defect density with efficient deep UV emission and lasing behavior at a photon energy of 5.7 eV (215 nm), as shown in Figure 1d, and claimed that multilayer hBN has a direct bandgap. The ultraviolet emission spectrum of multilayer hBN was acquired using cathodoluminescence (CL) spectroscopy, irradiating an electron beam on the sample to excite the valence band electrons and observing the light emission during the electron–hole recombination. Despite the efficient and intense emission from multilayer hBN around 5.7 eV (215 nm), the basic question of the nature of the bandgap properties and bandgap value of hBN was controversial. In 2016, G. Cassabois et al. presented evidence that hBN has an indirect bandgap, along with evidence of a phonon-assisted optical transition at 5.955 eV with 130 meV exciton-binding energy, through two-photon spectroscopy and temperature-dependent photoluminescence [9]. Nevertheless, hBN has emerged as a key material for the development of robust, next-generation optoelectronics due to its large bandgap (close to 6 eV, and even larger in some cases) and efficient phonon-assisted optical transition [10,11,12,13].

## 2. Color Centers in Ultrawide-Bandgap Semiconductors

In this chapter, we will look at the behaviors of defects and impurities in ultrawide-bandgap semiconductors, such as diamond and hBN. Single-crystal diamonds have a 5.47 eV bandgap, making them transparent in the visible wavelength range. However, occasionally, natural and artificial diamonds show various colors of luminescence depending on their natural vacancies or extrinsic impurities in the crystal lattice under ultraviolet excitation, as shown in Figure 2a [14]. Due to this phenomenon, optically active atomic defects are known as “color centers” in ultrawide-bandgap materials. Figure 2b shows various types of color centers—vacancy, substitutional, interstitial, and self-interstitial—in the crystal lattice [15]. A crystal can have a vacancy color center when the constructing atoms are evacuated. The vacant site can be substituted with other atoms to make a substitutional color center. The heterogeneous atom can be placed somewhere other than the exact crystal atom place, leading to an interstitial color center. When such a place is occupied by the original atom, it is called a self-interstitial color center.

In ultrawide-bandgap semiconductors like diamond and hBN, color centers give rise to stable energy states within the forbidden region of the bandgap of host materials, as shown in Figure 2c. The Franck–Condon principle explains the optical transition between the ground state and excited state of color centers [16]. According to the Franck–Condon principle, the nuclei are considered fixed due to their much larger mass compared to the electrons during an electronic transition. Let us think about a molecule consisting of two atoms. Initially, the electron is in a ground state, and two nuclei are placed at a certain distance, called optimal bound state position. If some amount of energy is given to the molecule, the electron jumps up from the ground state to the excited state. After the electron state has been changed, we see that the optimal bound state position of the nuclei (ν = 0) is no longer optimal with the new state of the electron. With the new optimal position being created, the nuclei start to move to the new optimal position, initiating the vibration of the nuclei. The same thing happens for the emission process. In short, the light absorption and emission produces molecules in vibrationally excited states (see Figure 2d). Now, we generalize this situation to a crystal consisting of many atoms. The degree of freedom of nuclear motion can lead to changes in the vibrational energy levels, taking into account the coupling between the electronic and vibrational modes. This explains the interplay between electronic and vibrational transitions in color centers in ultrawide-bandgap materials during processes such as the absorption, emission, or scattering of light.

**Figure 2 nanomaterials-13-02344-f002:**
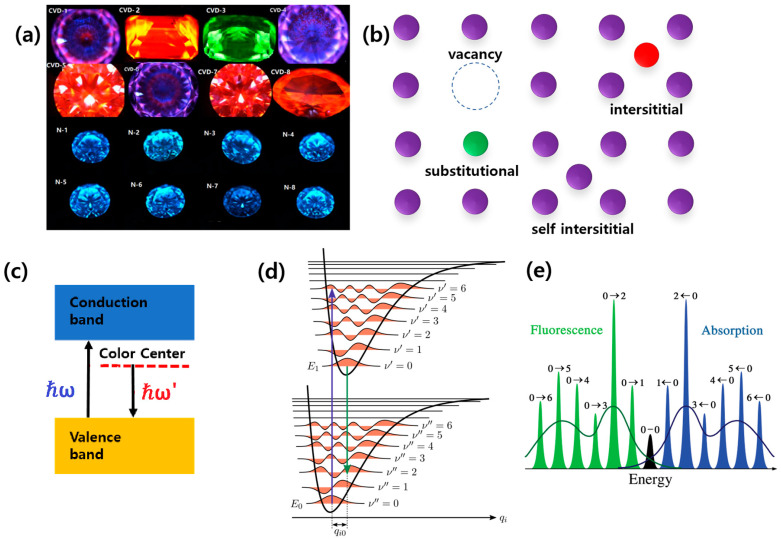
(**a**) Fabricated colored diamonds of various colors with luminescent characteristics. The colors were observed using ultraviolet fluorescence (265 nm) using DiamondView (De Beers, London, UK) [14]. (**b**) Types of color centers: vacancy, substitutional, interstitial, and self-interstitial color centers [15]. (**c**) The band diagram of the color center states formed in an ultrawide-bandgap semiconductor crystal. (**d**) A diagram of the Franck–Condon principle, showing the “ground state” and “excited state” with their own bound sub-states of the vibrational states. We see that the electron state change causes the new optimal position of the nuclei [16]. (**e**) Fluorescence and absorption spectrum of the Franck–Condon model, where the absorption spectrum is blue-shifted (higher energy) from the “energy gap”, while the emission spectrum is red-shifted (lower energy) from the energy gap. For example, the indication on the fluorescence peak 0→2 stands for the transition from ν’ = 0 to ν = 2.

The absorption and emission spectrum consists of a sharp line corresponding to an “energy gap” and other lower-energy lines, with the former spectral line being the zero-phonon line (ZPL), and the latter spectral lines being the phonon sideband (PSB) due to vibronic coupling. The separated spectral lines of the PSB are characteristic of an interband transition with a vibrational mode, such as optical and acoustic phonons, and the spectral wavelength of the ZPL represents the types of color centers. The principle further states that the intensity of the transition is directly proportional to the overlap between the wavefunctions of the initial and final electronic states. This overlap is determined by the relative positions of the nuclei in the two states involved in the optical transition of absorption and emission. In the color centers in ultrawide-bandgap materials, the Franck–Condon principle explains the optical transitions between the ground state and excited state of the color centers as well as their quantum properties (see Figure 2e).

Throughout the 2000s, significant progress was made in understanding and manipulating the quantum properties of NV center diamond, including quantum control of the electron spin, initialization, and readout techniques. [17] Despite the promising features, NV center diamond also has limitations. One of the main challenges is related to the coherence times, which can be affected by the surrounding environment and impurities in the crystal lattice. The scalability of NV center-diamond-based quantum devices has been hindered due to difficulties in fabricating large-scale devices and integrating them with other quantum components. In recent years, researchers have turned their attention to hBN as an alternative platform for quantum devices. hBN’s two-dimensional nature and unique crystal structure make it an attractive candidate for quantum technologies, especially for integration with quantum photonic circuits. As a result, efforts have been made to study color centers in hBN as a potential replacement for NV center diamond.

The band structure of diamond (Figure 3b) [18,19] is characterized by a large bandgap between the valence band and the conduction band. The NV center diamond introduces localized energy levels within the bandgap, creating an energy level scheme that involves electronic transitions between these levels. The NV center diamond has a ground state and two low-lying excited states, separated by a zero-phonon line (ZPL). The ZPL wavelength of the NV center in diamond is around 637 nm (nanometers) in the visible region of the electromagnetic spectrum. The ground state corresponds to the electronic configuration of a nitrogen atom in a substitutional site in a diamond lattice with an unpaired electron spin. The two excited states are associated with transitions involving the nitrogen spin and lattice vibrations (phonons).

The NV center diamond’s energy level structure exhibits spin-dependent optical transitions, meaning that its optical properties are sensitive to the spin state of the electron. This spin-dependent nature allows for an efficient and high-fidelity readout of the NV center’s electron spin state, a crucial property for quantum information processing. In conclusion, the NV center in diamond possesses unique properties related to its electronic bandgap, such as spin-dependent optical transitions and a sharp zero-phonon line emission peak [20]. Comparing NV center diamond and color centers in hexagonal boron nitride, hBN has a wider bandgap in the ultraviolet range, typically around 6 eV. The color centers in hBN exhibit broadband photoluminescence in the visible and UV regions. Diamond also has a large bandgap of approximately 5.5 eV, and the NV center diamond exhibits well-defined optical transitions with a zero-phonon line (ZPL) at around 637 nm in the visible spectrum.

The coherence times of the color centers in hBN are generally shorter than those of NV center diamond. Unlike carbon atoms in diamonds, nitrogen and boron atoms in hBN have nuclear spins, which hinder the spin from being in coherent states. On the contrary, NV center diamond are renowned for their long coherence times at room temperature. In Ref. [21], M. Ye et al. performed computer simulations to obtain the spin coherence times of four different 2D materials, namely delta-doped diamond layers, thin Si films, MoS_2_, and hBN. Compared to three other materials whose spin coherence times are around a few milliseconds, hBN exhibited significantly short spin coherence times, which are only about 10~30 microseconds, two orders of magnitude smaller than the others. The boron-vacancy color center in hBN (which will be discussed again in Section 5.2) is well known for its spin texture, but the spin coherence lifetime is limited compared to that of NV center diamonds. The experimental study Ref. [22] tells us that the spin coherence lifetime of an hBN defect measured via Rabi oscillation was 10 microseconds at a cryogenic temperature of T = 8 K, while an NV center diamond recorded a much longer spin coherence time of 400 microseconds even at room temperature.

However, improvements in coherence times have been shown by careful engineering of the local environment and isotopically purifying the hBN samples. The study Ref. [23] demonstrated that the coherent manipulation of V_B_^−^ spinful color centers in hBN was possible even at room temperature by applying pulsed spin resonance protocols. Moreover, at cryogenic temperature, spin-lattice relaxation time achieved the record of 18 microseconds, which is three orders of magnitude larger than its usual value. In Ref. [24], the authors performed computation on the temporal properties of decoherence by combining density functional theory (DFT) and cluster correlation expansion (CCE), and demonstrated that the coherence time can be extended by the factor of three by replacing all the boron atoms in the hBN crystal to ^10^B isotopes.

The luminosity factor of the color centers for quantum photonic applications is usually measured by the number of photons emitted from optically saturated single-photon emitters. In a previous study, a nitrogen-vacancy single-photon emitter [25] achieved 4.2 Mcps (million photon counts per second), showing compatibility with other materials, such as NV center diamond and SiC, with a brightness of roughly 0.1~1 Mcps.

## 3. Fabrication Process of Color Centers in hBN

### 3.1. Thermal Annealing Method

The thermal annealing method is a process that involves heating pristine hBN crystals to high temperatures, typically between 550 °C and 850 °C, in a vacuum and several gas environments, as shown in Figure 4a. This technique has the distinction of being the very first technique used to create quantum-light-emitting color centers in hBN crystals [25], and it has since become a common and standard method for producing single-photon sources at room temperature [26]. The color centers, such as intrinsic vacancies, are randomly generated through thermal annealing and undergo extensive examination using photoluminescence spectroscopy, revealing that a sharp zero-phonon line (ZPL) peak appears around 560–650 nm with multiple phonon sideband (PSB) peaks. It is worth noting that while the basic thermal annealing technique can produce color centers with sharp spectra and stable emissions, it does not necessarily ensure the stability of the spectrum or the deterministic wavelength and position of the color centers. Despite these limitations, thermal annealing remains a valuable technique for producing color centers in hBN crystals, and its widespread use has spurred the development of new and improved methods for color center fabrication.

### 3.2. UV Ozone Treatment Method

UV ozone treatment of pristine hBN is also used for color center activation. C. Li et al. demonstrated the creation of color centers and their single-photon emission from hBN through thermal annealing and UV ozone treatment [26]. Inside the UV ozone etcher, the ozone is produced from the oxygen molecules (O_2_) in the air. They are broken into individual oxygen atoms (O + O) by the high-power ultraviolet light from the UV lamp, which react with oxygen molecules to produce ozone molecules (O_3_). hBN samples are placed inside a commercially available UV ozone cleaner for 15, 30, and 60 min; then, the samples are examined via PL spectroscopy. Ozone-treated color centers in hBN have shown a sharp ZPL spectrum of 567.1 nm with an FWHM linewidth of 3.19 nm [26], which shows the advantage of the UV ozone treatment technique (see Figure 4b).

### 3.3. Laser Writing Method

The creation of deterministic atomic-scale vacancies or defects in an hBN crystal by firing a focused laser beam onto the sample was demonstrated by C. Palacios-Berraquero et al. [27]. They created a 2D lattice of color centers with a five-micrometer interval in hBN using a single-shot femtosecond pulse laser with a pulse width of less than 500 fs and an energy range from 30 to 60 nJ. Photoluminescence (PL) scanning of the sample revealed that the laser writing color center fabrication process is highly deterministic, with color center emissions (530–600 nm) visible from nearly all irradiated spots. Large-scale, high-yield lattice arrays of color center spots can be produced with this method. However, the ZPL wavelengths are widely distributed between 530 nm and 600 nm, rendering this technique only useful for independently operating photon sources (see Figure 4c).

### 3.4. Local Strain Method Using Micropillars

The transfer of hBN onto a nanofabricated pillar substrate shows the advantage of highly concentrated nanoscale strain on atomically thin crystals that can yield vacancies or defects. By utilizing a patterned substrate that generates a strained area via a nano-sized pillar, color centers can be created in intentional locations. This patterned substrate technique has also been applied to WSe_2_ monolayer crystals at cryogenic temperatures in the studies [28,29], and it has since been adapted to study room-temperature quantum emitters in hBN. In 2018, N. V. Proscia et al. [28] prepared an atomically thin hBN sheet grown through chemical vapor deposition (CVD) on a copper substrate. As depicted in Figure 4d [30], the authors mechanically transferred hBN onto a nanoscale pillar-patterned substrate to induce nano-sized strain at deterministic positions. Through this process, they were able to successfully generate color centers with an emission wavelength of approximately 540 nm at room temperature.

An alternative approach to the pillar substrate method involves growing hBN crystals natively strained at deterministic positions [31]. A thin hBN crystal can be fabricated using the CVD process by applying borazine gas onto a silicon oxide/silicon substrate at a high temperature of approximately 1200 °C. Unlike the previous method, this process intentionally creates defects early on during the crystal growth phase through natural crystal strain without the need for external atoms. To implement the pillar substrate method, one can prepare a nano-sized circular pillar array on the substrate using photolithography techniques. The pillars have a lattice constant of 2500 nm, a height of 650 nm, and a diameter of 500 nm. Through this preparation, CVD-fabricated thin hBN crystals naturally have strong strain on top of each pillar. Wide-field photoluminescence (PL) imaging reveals that at least half of the strain created on top of the pillars leads to bright emitting sites. The ZPL and PSB positions are around 608.5 and 663.9 nm, respectively, demonstrating a red visible emission. This method is particularly advantageous for producing color centers with deterministic positions.

**Figure 4 nanomaterials-13-02344-f004:**
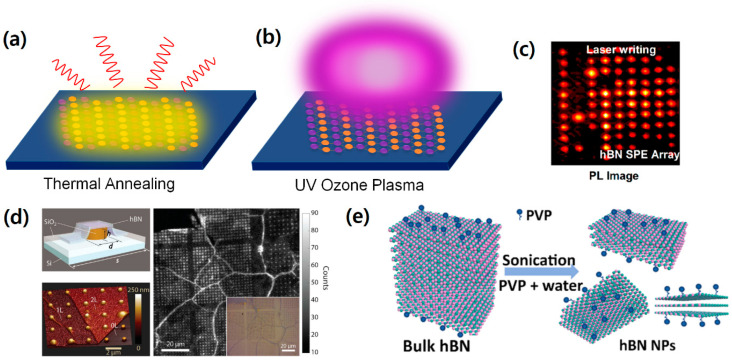
Fabrication methods of hBN color centers. (**a**) Thermal annealing method. The hBN crystals are usually placed inside a furnace at ~500 °C for ~30 min in a vacuum or environment to create defects in the crystal [26]. (**b**) UV ozone plasma method. The hBN crystals are placed inside a commercially available UV ozone cleaner for ~30 min, and the highly reactive ozone creates defects [26]. (**c**) Laser writing method. The hBN crystal is damaged with a single-pulse laser with energy of 50 nJ, and the array patterns are written [27]. [Reprinted with permission from L. Gan et al., “Large-Scale, High-Yield Laser Fabrication of Bright and Pure Single-Photon Emitters at Room Temperature in Hexagonal Boron Nitride”, ACS Nano, 16, 14254–14261 (2022), Copyright 2022, American Chemical Society]. (**d**) Pillar array method. A thin layer of hexagonal boron nitride is transferred to silicon oxide with a micro-size pillar array to induce local strain points in the hBN layer. The characteristic emissions are observed using confocal PL mapping on every single pillar point, showing the near-deterministic nature and high brightness of the pillar emitters [30]. [Reprinted with permission from N. V. Proscia et al., “Near-deterministic activation of room-temperature quantum emitters in hexagonal boron nitride”, Optica, 5, 1128 (2018), Copyright 2018, American Chemical Society]. (**e**) Solvent exfoliation method. The hBN crystal is originally hydrophobic, but surface treatment with polyvinylpyrrolidone (PVP) molecules makes hBN soluble in water [32]. [Reprinted with permission from Y. Chen et al., “Solvent-Exfoliated Hexagonal Boron Nitride Nanoflakes for Quantum Emitters”, ACS Applied Nano Materials, 4, 10449–10457 (2021), Copyright 2021, American Chemical Society].

### 3.5. Solution Exfoliation Method

The natural hBN crystal possesses hydrophobic properties and is insoluble in water or any other polar solvent. However, to make it soluble, a water-soluble polymer-based surfactant, polyvinylpyrrolidone (PVP), can be added to the solution. PVP has a large molecular chain of 40,000 Da and is commonly used in nanoparticle synthesis and biomedical research. It reduces the surface energy of water, making hBN powder soluble in water. To exfoliate thin hBN nanoflakes, hBN powder is added to an aqueous PVP (0.1 M) solution, and a probe ultra-sonicator is submerged into the solution for half an hour. The resulting water-dissolved hBN solution can then be dropped onto a thermal oxide silicon substrate. The substrate is then annealed at 850 °C in argon (1 Torr) to completely dry the solution [32]. The aqueous hBN solution is applied to the silicon substrate and completely dried to leave hBN flakes with random natural defects and color centers. This process not only produces an hBN crystal but also activates color centers for use as light emitters. Furthermore, this annealing technique can also be used to make a pristine hBN crystal defective. The resulting crystal from the heat-dried solution now has defects that can be used as a color center light emitter. Although the distribution of wavelengths is not very homogeneous, with values ranging from 582 nm to 614 nm in five different samples, this technique still produces useful and independently acting color center emitters (see Figure 4e).

## 4. Wavelength of Color Centers in hBN

The emissions of photons with different wavelengths depend on the types of color centers in hBN, and they display a stable ZPL emission with single-photon emission and spin qubit characteristics at room temperature. These properties of color centers in hBN make them promising candidates for solid-state UV light sources and quantum information applications. Optical excitation and PL spectroscopy observations are the most commonly used and easily accessible methods to study the electronic structure of hBN color centers. Recent PL spectroscopy studies have revealed the existence of numerous color centers in hBN, making it crucial to survey observed color center emissions to identify the origin of the atomic configuration of color centers in hBN.

In this paper, we compiled data from recent research papers published between the years 2016 and 2023 ([6,7,8,9,10,11,12,13,14,15,16,17,18,19,20,21,22,23,24,25,26,27,28,29,30,31,32,33,34,35,36,37,38,39,40,41,42,43,44,45,46,47,48,49,50,51,52,53,54,55,56,57,58,59,60,61,62,63,64,65,66,67,68,69,70,71,72,73,74,75,76,77,78,79,80,81,82,83,84,85,86,87,88,89,90,91,92,93,94,95,96,97,98,99]). For further analysis, we present a histogram of the ZPL wavelength of the color centers in hBN, as shown in Figure 5a, which includes 148 data points. The results show that the majority (nearly 50%) of the ZPL is located in the green-to-red visible range (550~650 nm), with some (about 15%) ultraviolet emissions (300~400 nm) and a small number of near-infrared (near-IR) emissions longer than 750 nm. To represent each region of the wavelength, we selected three representative emission spectra, which are shown in Figure 5b–d. Figure 5b from Ref. [39] represents a rare deep ultraviolet emission (~303 nm) from a carbon substitutional color center at a nitrogen site, excited via cathodoluminescence (CL). Figure 5c from Ref. [25] represents the green-to-red visible region (~630 nm), where the majority of hBN color centers are located. Finally, Figure 5d from Ref. [22] represents the near-infrared (near-IR) region (~850 nm) from a boron-vacancy color center. Based on this survey, we can conclude that hBN color center emission covers almost all ranges of visible light, including the deep UV and near-IR regions.

## 5. Atomic Configuration of Color Centers in hBN

In the previous chapter, we explored the optical characteristics of color centers in hBN. However, to gain a deeper understanding of the origins of these optical responses, we need to reveal the specific color center types within the atomic-scale microscopic structures. Investigating the source of luminescence in color centers in hBN proves to be challenging, but several potential candidates have been suggested. Figure 6 presents a possible illustration of the atomic configuration of diverse hBN color centers with vacancies and substitute carbon atoms.

### 5.1. Nitrogen Vacancy

The generation of nitrogen-vacancy color centers in hBN crystals is commonly achieved through the argon-atmosphere thermal annealing method. By employing ab initio computations, the energy diagrams for the two distinct types of nitrogen-vacancy color centers have been studied. These color centers have been experimentally observed using optical spectroscopy and single-photon emission experiments, as depicted in Figure 7 [25]. Figure 7a shows a room-temperature PL spectroscopy result, which compares the spectra of color centers from multilayer and monolayer hBN. Both spectra have the same sample ZPL peak wavelength of 625 nm (1.98 eV). However, the line widths of the spectra are different. The multilayer color center has a narrow emission with a full width at half maximum (FWHM) of around 5 nm, which allows for easily distinguishing between the ZPL peak (a single peak at 625 nm) and the PSB peaks (double peak around 680 nm). On the other hand, the monolayer color center has a broad emission with an FWHM of around 20 nm, which only allows for the observation of the ZPL peak. We attribute the broader phonon sideband emission from the monolayer hBN to the stronger phonon interaction with the substrate in monolayer hBN crystal structure, where the atomically thin monolayer hBN is more vulnerable to electron–phonon interaction.

The study of reference [25] proposed the atomic structure of the color center between the ordinary nitrogen vacancy (V_N_) and the anti-site nitrogen vacancy (N_B_V_N_), which is a nitrogen-vacancy color center where boron is substituted for nitrogen. Figure 7b shows the atomic structure of N_B_V_N_. This particular N_B_V_N_ configuration is a common form of nitrogen-vacancy color center in hBN, with bright emission at a photon energy of 1.9~2.15 eV and stable quantum emission properties at room temperature. The quantum emission properties of N_B_V_N_ were confirmed to be PL intensity saturation features and single-photon emission through a second-order time correlation g^(2)^ measurement, as shown in Figure 7c,d. In order to distinguish between the two types of nitrogen-vacancy color centers, V_N_ and N_B_V_N_, it is necessary to consider not only the difference in the ZPL wavelength but also the polarization characteristics. Unlike the color center type V_N_, which exhibits azimuthal rotation with a 360° rotational symmetric emission, the color center type N_B_V_N_ emits highly anisotropic and linearly polarized photons. The study of Ref. [25] observed a highly linearly polarized emission, which indicated that the color center corresponded to N_B_V_N_ based on its polarization properties and characteristic wavelength of 630 nm.

**Figure 7 nanomaterials-13-02344-f007:**
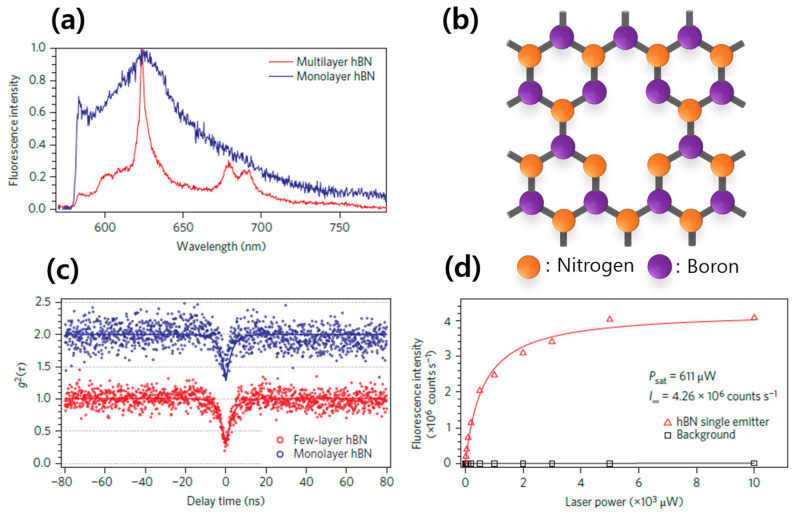
Nitrogen-vacancy-based hBN color center [25]. [Reprinted with permission from T. T. Tran et al., “Quantum emission from hexagonal boron nitride monolayers”, Nature Nanotech, 11, 37–41 (2016)]. (**a**) Room-temperature photoluminescence spectra of a defect center in an hBN monolayer (blue) and multilayer (red). (**b**) Schematics of the anti-site nitrogen vacancy N_B_V_N_. (**c**) Antibunching curves from an individual defect center in an hBN monolayer (blue open circles) and multilayer (red open circles). (**d**) Fluorescence saturation curve obtained from a single defect.

### 5.2. Boron Vacancy

The boron-vacancy color center in hBN usually appears in the form of a negatively charged state of V_B_^−^, as shown in Figure 8a, which is often created by bombarding high-energy particles, such as heavy ions, neutrons, and electrons. PL spectroscopy of the V_B_^−^ color center generated by bombarding a lithium/gallium ion beam showed a characteristic broad peak centered at around 850 nm [22], as shown in Figure 8b. The zero external magnetic field energy level diagram of V_B_^−^ is well known to be accompanied by an electronic spin-triplet (S = 1) system (see Figure 8c). The three triplet states are grouped into two levels, m_S_ = 0 and m_S_ = ±1, where the two states of m_S_ = ±1 are energetically degenerated.

The important factor describing the negatively charged boron-vacancy color center in hBN, V_B_^−^, is the zero-field splitting (ZFS) splitting D value, which is the energy splitting between the state m_S_ = 0 and the states m_S_ = ±1 (degenerated, same values). The value of ZFS is D = 14 μeV, which corresponds to D/h~3.5 GHz. This kind of ZFS can be detected through optically detected magnetic resonance (ODMR) measurements, as shown in Figure 8d, and has been found to remain stable even in high-temperature environments up to 600 K. Under an external magnetic field applied to the V_B_^−^ color center, ZFS level splitting between the states of m_S_ = ±1 is increased, and magnetic-field-dependent ODMR splitting can be measured, as shown in Figure 8c,d [22,85,86]. An electron paramagnetic resonance (EPR) measurement can be performed to directly confirm that the emission is from the boron vacancy by detecting a seven-line structure induced by the hyperfine interaction of the electron spin with three equivalent nitrogen-14 nuclei. The zero-field splitting of V_B_^−^ varies sensitively with the temperature, pressure, external magnetic field, and strain. For example, the ZFS value of V_B_^−^ varies by ~120 MHz between room temperature and 4 K out of 3.5 GHz. The V_B_^−^ color center instability can be exploited to detect any change in the environment, paving the way for quantum sensors.

**Figure 8 nanomaterials-13-02344-f008:**
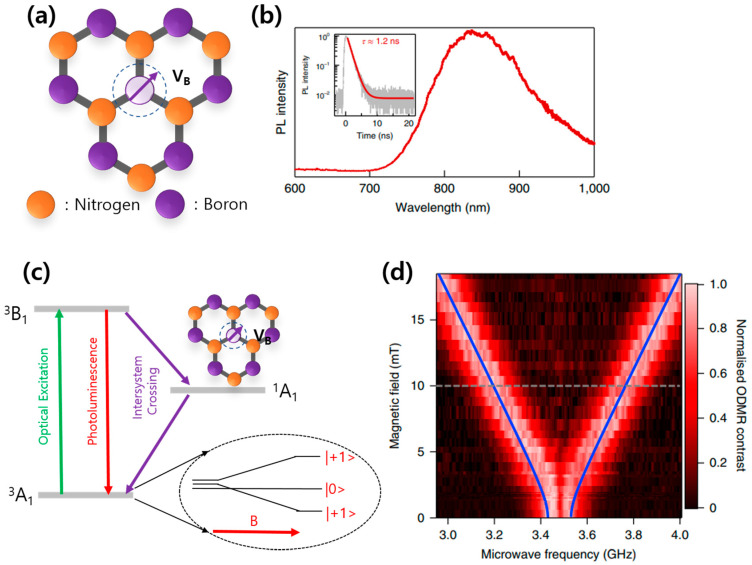
Boron-vacancy color center in hBN. (**a**) Schematic crystal structure of a negatively charged boron-vacancy color center. (**b**) PL spectroscopy result of V_B_^−^ showing the characteristic broad emission spectrum centered around 850 nm. (**c**) Energy diagram of V_B_^−^ color center energy levels depicting the triplet ground state. (**d**) Dependence of ODMR frequencies as a function of the magnetic field (B || c) [22]. [Reprinted with permission from A. Gottscholl et al., “Initialization and read-out of intrinsic spin defects in a van der Waals crystal at room temperature”, Nature Materials, 19, 540–545 (2020)].

### 5.3. Oxygen Impurities

Oxygen vacancies in hBN crystals are typically created through thermal annealing or argon plasma etching. In a study by Yang et al. [86], a PL spectrum with an emission peak at 711 nm was observed at room temperature and 11 K after argon plasma etching and annealing of hBN crystals. A PL mapping comparison also showed the effect of oxygen-plasma-induced color center creation. X-ray photoemission spectroscopy (XPS) confirmed the presence of oxygen bonding after plasma etching, indicating the existence of an oxygen-impurity-based hBN color center. Among the theoretically possible oxygen color centers, V_B_O_2_, a two-oxygen-atom hBN color center, is proposed as a strong candidate for luminescent oxygen color centers. Another method to create oxygen-related hBN color centers is oxygen plasma etching. In 2021, Na et al. [87] demonstrated how to modulate hBN’s optical and electrical properties by inducing color centers via oxygen plasma treatment. Their study showed prominent PL peaks around 720 nm, and as the time of exposure to the oxygen plasma became longer, the characteristic PL peaks of the oxygen color centers O_N_, V_B_, V_B_O_N_, V_B_O_2_, and V_N_ also grew (Figure 9).

### 5.4. Carbon Impurities

We observed that the color center types mentioned earlier cover a significant portion of the visible wavelength range. The nitrogen-vacancy color center spans approximately 600 nm, the oxygen impurity color center covers around 700 nm, and the boron-vacancy color center extends to approximately 850 nm. Now, we will shift our focus to a new wavelength region, namely the blue and UV emitters, with wavelengths ranging from 300 to 450 nm. Carbon-related color centers are usually created by injected carbon impurities during the synthesis process of the hBN crystal itself. The synthesis process of hBN is usually carried out in a high-pressure high-temperature (HPHT) environment or an atmospheric-pressure high-temperature (APHT) environment, leading to a significant amount of impurity injection, and carbon is the impurity with the largest portion.

As demonstrated in Figure 1b, CL spectroscopy of single-crystal high-purity pristine hBN crystals does not exhibit any discernible peak in the vicinity of approximately 4.1 eV (around 300 nm). However, Figure 10a [89] reveals that hBN crystals enriched with carbon display a distinct and vibrant emission at approximately 4.1 eV (300 nm). A comparison of the spectra between high-purity pristine hBN and carbon-rich hBN illustrates that the addition of carbon impurities significantly suppresses the prominent 215 nm peak observed in the pure sample by one order of magnitude. Instead, new peaks emerge in the range of 300 nm to 350 nm (corresponding to photon energies of 3.5 to 4.1 eV), revealing the presence of a deep ultraviolet (UV) optical gap within the hBN bandgap.

The vibrational peaks observed in the PL, in accordance with the Franck–Condon principle discussed in Section 2, are noteworthy. In a prior investigation [90], A. Vokhmintsev et al. conducted a detailed analysis of a well-defined PL spectrum associated with the carbon-impurity-related color centers in hBN. These clear vibrational peaks show a significant sign of quantum oscillation inside the carbon color center. This luminescence is suspected to originate from carbon dimers like C_N_C_B_ or carbon monomer defects like C_B_ and C_N_. A carbon atom can form not only these simple defect structures but also complex color centers such as V_B_C_N_ or V_N_C_B_. Even more complex structures are possible, as shown in Figure 1c [91], with the extended set of hBN carbon color centers with different numbers of substituted carbon atoms; i.e., 2, 4, and 6. Figure 10b shows a simulated PL spectrum of carbon color centers in the form of a 6C ring, 4C pair, and 2C carbon dimer through ab initio computation, which matches well with the experimentally measured PL spectroscopy of carbon impurities in hBN.

There have also been computational studies on V_B_C_N_ and V_N_C_B_ for platforms of spin–orbit coupling, optical transition, and ZFS. V_B_C_N_ color centers show visible luminescence and hold a triplet ground state, while V_N_C_B_ holds a singlet ground state. The spin texture of these carbon color centers is also under investigation for making use of them for quantum memory. Carbon-based color center defects in hBN are being actively studied as single-photon emitter platforms for quantum information; in particular, the stable emission of an ultraviolet photon from the 6C color center is gathering the most attention ([91,92,93,94,95,96,97,98,99]).

Recently, there has been prominent research on carbon-based color centers, which are called blue emitters (see Figure 10d,e [99]). In one study, color centers were created using the electron beams of a commercial SEM, and the low-temperature (down to 5 K) spectroscopy study revealed that the emission from this type of color center has a narrow emission wavelength of 435.5 ± 0.3 nm. The great repeatability and single-photon purity of the blue emitter technique show its potential for creating the ideal quantum light source, paving the way for optical quantum information usage.

## 6. Conclusions

Through a comprehensive review of the color centers in hBN, we explored various aspects of hBN color centers, encompassing their fundamental photon emission principles, categorization based on emitted wavelengths, fabrication methods, and microscopic atomic structures. We summarized the unique properties associated with each type of defect, providing insights into their specific characteristics and potential applications. The literature survey revealed that due to the ultrawide bandgap of hBN, color centers can exhibit a diverse range of colors spanning the visible, near-infrared, and ultraviolet regions. Among the atomic vacancies and impurities, the carbon color centers stand out as promising candidates. The carbon color centers demonstrate theoretically predictable room-temperature UV emission characterized by remarkable brightness and stability. This aspect renders them highly intriguing for potential applications in quantum technologies and advanced UV optoelectronics.

Some of hBN’s key applications include, but are not limited to, quantum photonics and UV optoelectronics. In quantum photonics, hBN is being explored as a platform for on-chip integrated quantum photonic devices. It can be used to create sources of single photons from color centers, which are crucial for quantum information processing and quantum key distribution. Compared to other materials, such as NV center diamond and silicon carbide (SiC), hBN has a great advantage in that it is an atomically thin 2D material; therefore, its integration into quantum photonic chips and the manipulation of optical properties, such as straining the device, are much easier.

In UV optoelectronics, hBN possesses a wide bandgap, making it an excellent candidate for UV optoelectronic applications. It can be used to create efficient UV light emitters, detectors, and sensors. hBN-based LEDs can be used in advanced UV lighting applications, such as sterilization, water purification, and UV curing processes, in industries. These applications highlight the broad potential of hBN in advancing quantum technologies and UV optoelectronics, enabling the development of more efficient, compact, and robust devices for various scientific and industrial applications.

## Figures and Tables

**Figure 1 nanomaterials-13-02344-f001:**
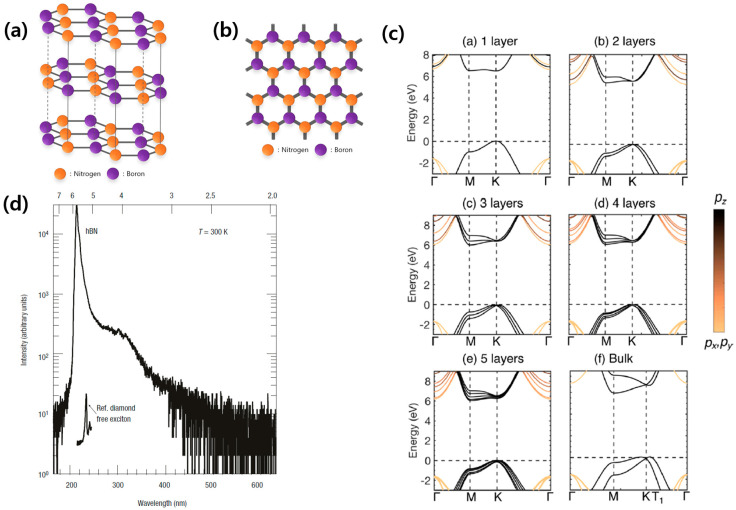
(**a**) Schematic diagram of the hBN crystal structure, showing the hexagonal bonding structure in each plane and the AA′ interlayer bonding structure, with orange balls representing nitrogen atoms and purple balls representing boron atoms. (**b**) Schematic diagram of each unit layer of the hBN crystal, seen from the top view, perpendicular to the plane. (**c**) (a–f) Band structure diagrams of hBN with the number of atomic stacked layers being (a) 1, (b) 2, (c) 3, (d) 4, (e) 5, and (f) bulk. As the layer number increases, the material evolves from a direct to an indirect bandgap material [7]. [Reprinted with permission from D. Wickramaratne et al., “Monolayer to Bulk Properties of Hexagonal Boron Nitride”, Journal of Physical Chemistry C, 122, 25524–25529 (2018). Copyright 2018, American Chemical Society]. (**d**) A cathodoluminescence spectrum of high-purity single-crystal hBN. A narrow peak appears at the photon energy position 5.8 eV (215 nm), showing the deep UV optical gap of the hBN crystal [8]. [Reprinted with permission from K. Watanabe et al., “Direct-bandgap properties and evidence for ultraviolet lasing of hexagonal boron nitride single crystal”, Nature Materials, 3, 404–409 (2004)].

**Figure 3 nanomaterials-13-02344-f003:**
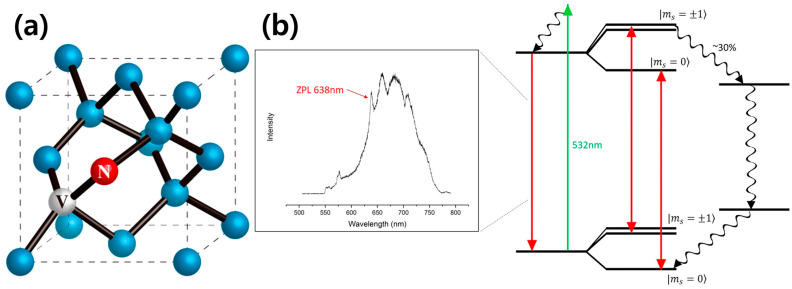
(**a**) Simplified atomic structure of the NV center diamond, consisting of a substitutional nitrogen atom (red), an atomic vacancy (white), and carbon atoms (blue) [17]. (**b**) PL spectrum and energy level diagram in the NV center in diamond. The primary transition between the ground- and excited-state triplets is spin conservation. Decay via the intermediate singlets gives rise to spin polarization by converting spin from ms = ±1 to ms = 0 [18].

**Figure 5 nanomaterials-13-02344-f005:**
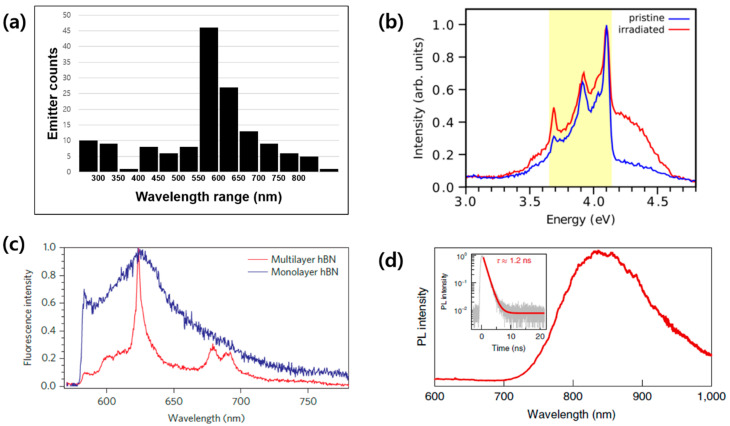
A survey of hBN color center emission wavelengths. (**a**) ZPL wavelength of color centers in the hBN distribution [6,7,8,9,10,11,12,13,14,15,16,17,18,19,20,21,22,23,24,25,26,27,28,29,30,31,32,33,34,35,36,37,38,39,40,41,42,43,44,45,46,47,48,49,50,51,52,53,54,55,56,57,58,59,60,61,62,63,64,65,66,67,68,69,70,71,72,73,74,75,76,77,78,79,80,81,82,83,84,85,86,87,88,89,90,91,92,93,94,95,96,97,98,99]. (**b**) Deep ultraviolet emission with a photon energy of 4.1 eV (equivalent to the wavelength of 303 nm) from a carbon impurity color center [35]. [Reprinted with permission from R. Bourrellier et al., “Bright UV Single Photon Emission at Point Defects in h-BN”, Nano Letters, 16, 4317–4321 (2016), Copyright 2016, American Chemical Society]. (**c**) Visible emission of 630 nm from a nitrogen-vacancy color center [25]. [Reprinted with permission from T. T. Tran et al., “Quantum emission from hexagonal boron nitride monolayers”, Nature Nanotech, 11, 37–41 (2016)]. (**d**) Near-infrared emission of 850 nm from boron-vacancy color centers [22]. [Reprinted with permission from A. Gottscholl et al., “Initialization and read-out of intrinsic spin defects in a van der Waals crystal at room temperature”, Nature Materials, 19, 540–545 (2020)].

**Figure 6 nanomaterials-13-02344-f006:**
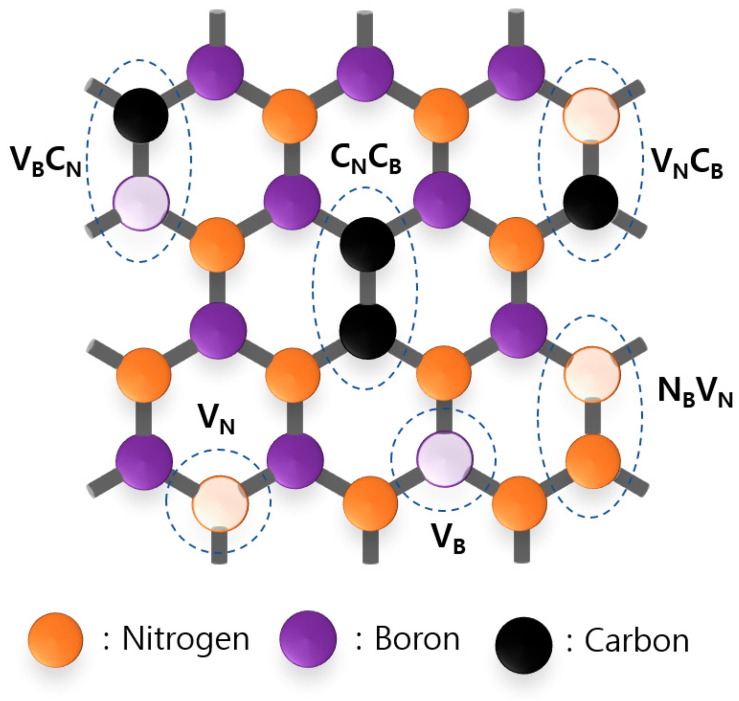
Atomic structure of color centers in hBN with intrinsic vacancies and carbon impurities. There are various hBN color center types with substitute atoms. Here, we present V_N_ (nitrogen vacancy), V_B_ (boron vacancy), V_B_C_N_ (boron vacancy and carbon substitution at the nitrogen site), V_N_C_B_ (nitrogen vacancy and carbon substitution at the boron site), and C_B_C_N_ (carbon substitutions at the nitrogen and boron sites).

**Figure 9 nanomaterials-13-02344-f009:**
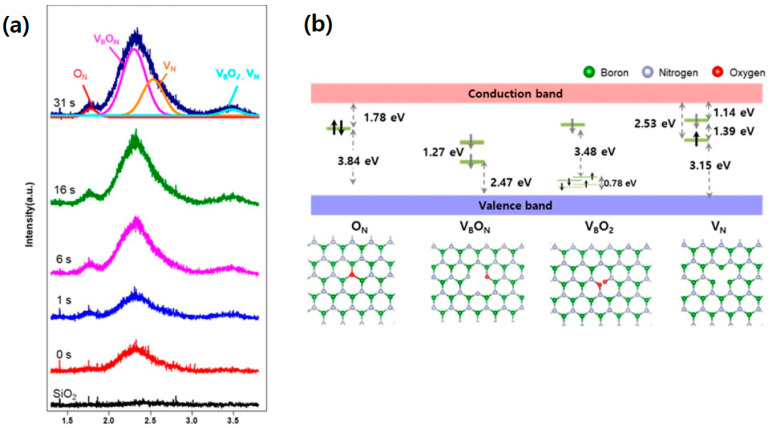
Oxygen-impurity-based hBN color center. (**a**) Photoluminescence (PL) spectra of hBN, with the oxygen plasma treatment time. (**b**) Color center structures of O_N_, V_B_, V_B_O_N_, V_B_O_2_, and V_N_, and electronic structure. Black arrows indicate spin-up/down in occupied states, while gray arrows indicate empty states [87]. [Reprinted with permission from Y. Na et al., “Modulation of optical and electrical properties in hexagonal boron nitride by defects induced via oxygen plasma treatment”, 2D Materials, 8, 045041 (2021)].

**Figure 10 nanomaterials-13-02344-f010:**
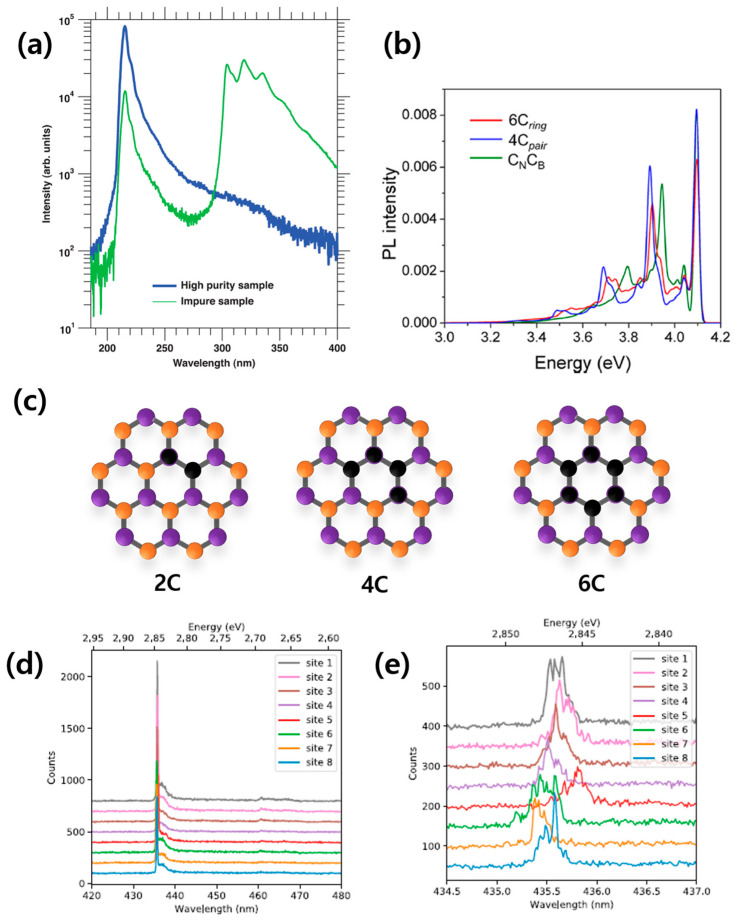
Carbon-impurity-based hBN color center. (**a**) Cathodoluminescence spectrum of a high-purity hBN sample and an impure sample. As the impurity is added, the 215 nm peak is quenched by an order of magnitude, and other peaks appear around 300 nm~350 nm, showing the deep UV optical gap of the hBN crystal [89]. (**b**) Simulated PL spectrum of a dimer (C_N_C_B_), 4C pair, and 6C ring where the ZPL energies are aligned for the sake of comparison of PSBs [90]. (**c**) Extended set of hBN carbon color centers with different numbers of substituted carbon atoms: 2, 4, and 6. Orange: nitrogen; purple: boron; black: carbon [91]. (**d**,**e**) Low-temperature (5 K) spectra of the eight spots with two different spectral resolutions showing a reproducible ZPL within 0.7 nm [99].

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
