# Peer review of "Color Centers in Hexagonal Boron Nitride"

_nanomaterials, 2023, doi:10.3390/nano13162344_

Round 1

Reviewer 1 Report

This manuscript reviews the color center of hexagonal boron nitride, which have been widely studied in recent years. The color centers of hexagonal boron nitride are expected to be one of optical materials for next-generation quantum electronics such as single-photon sources and quantum sensor applications, just like the NV center in diamond. In recent years, this type of research papers has been extensively reported, and the field is advancing rapidly. Thus, the comprehensive review like this manuscript is very useful and valuable for many readers.

This paper covers basic information such as the crystal structure and the band structure of hexagonal boron nitride, the method of forming various color centers, and the types of defect structures in color centers. They are very concise and easy to look over the recent research. However, as I mentioned above, this field is advancing rapidly day by day, and some important and essential topics for quantum aspects are missing. The quantum color center research on hexagonal boron nitride to date can be summarized step by step, as follows.

 Stage 1: Formation of color centers and their properties as a single light source.

 Stage 2: Establishing methods for generating color centers and confirming the feasibility of quantum applications such as spin operations.

 Stage 3: Deliberating properties necessary for practical quantum applications, such as luminous efficiency and spin relaxation time for color centers.

 Stage 4: Proposal and demonstration of concrete quantum device development

Many advanced researchers in this field are now interested in the third or fourth stages based on the second stage. However, the scope of this paper is unfortunately limited to the second stage. Therefore, I would like to suggest the authors to describe the following additional topics.

(1) "Chap 2. Color centers in ultra-wide band gap semiconductor"

This chapter briefly explains the fundamentals of physics in defect luminescence giving an example of diamond color centers, but lacks the description of the diamond NV center, the pioneer to the research for quantum application of defect luminescence in history. The comparison with this existing center is very important. Therefore, adding the description of the NV center in this chapter, then the authors should discuss the features of the hexagonal boron nitride color center compared to that of NV center through this paper.

(2) A description of coherent time, such as spin relaxation time, should be added.

Unlike diamonds made of carbon, nitrogen and boron have nuclear spins, so there is a question that the spin coherent states do not last long. For example, as discussed in the following paper, it is necessary to add a review of coherent time as a comparison of diamonds and other materials such as SiC.

Ye, M., Seo, H. & Galli, G. ,npj Comp.Mater. 5, 44 (2019).

Gottscholl, A. et al., Nat. Mater. 19, 540–545 (2020).(Ref.79)

Gottscholl, A. et al., Sci. Adv. 7, eabf3630 (2021).

Jaewook Lee et al., npj 2D Materials and Applications, (2022)6:60 

(3) Luminous efficiency: Already, as shown in the Ref. 19, specific luminous efficiency magnitudes have been discussed in several papers. Since luminous efficiency is very important for the applications, it is necessary to discuss the efficiency of each color center as quantitative as possible.

(4) Finally, the future prospects for the quantum application of the hBN color center should be discussed by comparing it with the color centers of other materials. This will attract the interest of many readers in the research of the hexagonal boron nitride color center. 

Minor correction

Page 1 line36 graphite -> graphene

Since the crystal structure of a single-layer hexagonal boron nitride is described here, graphene (monolayer) should be used instead of graphite (multi layers).

Author Response

Comment from Reviewer 1. and the Author’s Response.

Comment 0.

This manuscript reviews the color center of hexagonal boron nitride, which have been widely studied in recent years. The color centers of hexagonal boron nitride are expected to be one of optical materials for next-generation quantum electronics such as single-photon sources and quantum sensor applications, just like the NV center in diamond. In recent years, this type of research papers has been extensively reported, and the field is advancing rapidly. Thus, the comprehensive review like this manuscript is very useful and valuable for many readers.

Thank you very much for your positive review. Since the research area on color centers in hexagonal boron nitride is rapidly growing field, we consider a thorough review at this moment is very important. Once again, we thank to the Reviewer 1 for the positive consideration to our review work.

This paper covers basic information such as the crystal structure and the band structure of hexagonal boron nitride, the method of forming various color centers, and the types of defect structures in color centers. They are very concise and easy to look over the recent research. However, as I mentioned above, this field is advancing rapidly day by day, and some important and essential topics for quantum aspects are missing. The quantum color center research on hexagonal boron nitride to date can be summarized step by step, as follows.

Thank you very much for understanding the structure of our review paper and positive valuation.

 Stage 1: Formation of color centers and their properties as a single light source.

 Stage 2: Establishing methods for generating color centers and confirming the feasibility of quantum applications such as spin operations.

 Stage 3: Deliberating properties necessary for practical quantum applications, such as luminous efficiency and spin relaxation time for color centers.

 Stage 4: Proposal and demonstration of concrete quantum device development

Many advanced researchers in this field are now interested in the third or fourth stages based on the second stage. However, the scope of this paper is unfortunately limited to the second stage. Therefore, I would like to suggest the authors to describe the following additional topics.

Thank you very much for reminding us the recent progress of hBN color center research. We totally agree with Reviewer 1’s opinion that it will be valuable to compare hBN and color centers of other materials. However, since the detail origins of hBN color center emissions are still unexplored and under investigation, we concentrated on the color center study of hBN itself in our review paper. we decided to review and summarize the wide range of hBN color center emission wavelength first, to widen our scope to the Stage 3, 4 that the Reviewer 1 mentioned. We added some amount of consideration on Stage 3, 4. Please refer our Response below.

Comment 1.

"Chapter 2. Color centers in ultra-wide band gap semiconductor"

This chapter briefly explains the fundamentals of physics in defect luminescence giving an example of diamond color centers, but lacks the description of the diamond NV center, the pioneer to the research for quantum application of defect luminescence in history. The comparison with this existing center is very important. Therefore, adding the description of the NV center in this chapter, then the authors should discuss the features of the hexagonal boron nitride color center compared to that of NV center through this paper.

Response 1.

Thank you for providing these insights. In Chapter 2. “Color centers in ultra-wide bandgap semiconductor”, in order to investigate the hBN color center, we first wanted to explain the theory of color centers in general semiconductor with wide bandgap, not only limited to hBN. We added the three references about NV center diamond and numbered them to be [18], [19], and [20], as follows.

  1. F. Alghannam et al.,"Engineering of Shallow Layers of Nitrogen Vacancy Colour Centres in Diamond Using Plasma Im-mersion Ion Implantation", Scientific Reports, 9, 5870 (2019)
  2. A. Gruber et al., “Scanning confocal optical microscopy and magnetic resonance on single defect centers”, Science, 276, 2012– 14 (1997)
  3. H. Ariful et al., “An Overview on the Formation and Processing of Nitrogen-Vacancy Photonic Centers in Diamond by Ion Implantation”, J. Manuf. Mater. Process., 1(1), 6 (2017)

In the revised version of the manuscript, we inserted some paragraphs explaining the properties of diamond and comparison with hBN. The inserted paragraphs are followings, in page 5 ~ 6.

“In order to provide specific information, we compare nitrogen-vacancy centers in diamond with the color centers in hexagonal boron nitride. The research on quantum de-vices using nitrogen-vacancy (NV) centers in diamond (Fig 3. (a)) [17] dates back to the late 1990s when NV centers were first observed and characterized. The NV center in diamond is a point defect composed of a substitutional nitrogen atom adjacent to a vacant lattice site (vacancy). The detection of electron paramagnetic resonance (EPR) from a single NV center led to the study of quantum devices. Researchers discovered that NV centers possess unique properties, such as long electron spin coherence times even at room temperature, making them promising candidates for quantum information processing and sensing applications.

Throughout the 2000s, significant progress was made in understanding and manipulating the quantum properties of NV center diamond, including quantum control of the electron spin, initialization, and readout techniques. Despite the promising features, NV-center diamond also has limitations. One of the main challenges is related to the coherence times, which can be affected by the surrounding environment and impurities in the crystal lattice. The scalability of NV-center-diamond-based quantum devices has been hindered due to difficulties in fabricating large-scale devices and integrating them with other quantum components. In recent years, researchers have turned their attention to hBN as an alternative platform for quantum devices. hBN's two-dimensional nature and unique crystal structure make it an attractive candidate for quantum technologies, especially for integration with quantum photonic circuits. As a result, efforts have been made to study color centers in hBN as a potential replacement for NV centers in diamond.

The band structure of diamond (Fig 3. (b)) [18] [19] is characterized by a large bandgap between the valence band and the conduction band. The NV center diamond introduces localized energy levels within the bandgap, creating an energy level scheme that involves electronic transitions between these levels. The NV center diamond has a ground state and two low-lying excited states, separated by a zero-phonon line (ZPL). The ZPL wavelength of the NV center in diamond is around 637 nm (nanometers) in the visible region of the electromagnetic spectrum. The ground state corresponds to the electronic configuration of a nitrogen atom in a substitutional site in a diamond lattice with an unpaired electron spin. The two excited states are associated with transitions involving the nitrogen spin and lattice vibrations (phonons).

The NV center diamond's energy level structure exhibits spin-dependent optical transitions, meaning that its optical properties are sensitive to the spin state of the electron. This spin-dependent nature allows for an efficient and high-fidelity readout of the NV center's electron spin state, a crucial property for quantum information processing. In conclusion, the NV center in diamond possesses unique properties related to its electronic bandgap, such as spin-dependent optical transitions and a sharp zero-phonon line emission peak. [20] Comparing NV centers in diamond and color centers in hexagonal boron nitride, hBN has a wider bandgap in the ultraviolet range, typically around 6 eV. The color centers in hBN exhibit broadband photoluminescence in the visible and UV regions. Diamond also has a large bandgap of approximately 5.5 eV, and the NV center diamond exhibits well-defined optical transitions with a zero-phonon line (ZPL) at around 637 nm in the visible spectrum.

Comment 2.

A description of coherent time, such as spin relaxation time, should be added.

Unlike diamonds made of carbon, nitrogen and boron have nuclear spins, so there is a question that the spin coherent states do not last long. For example, as discussed in the following paper, it is necessary to add a review of coherent time as a comparison of diamonds and other materials such as SiC.

Ye, M., Seo, H. & Galli, G. ,npj Comp.Mater. 5, 44 (2019).

Gottscholl, A. et al., Nat. Mater. 19, 540–545 (2020).(Ref.79)

Gottscholl, A. et al., Sci. Adv. 7, eabf3630 (2021).

Jaewook Lee et al., npj 2D Materials and Applications, (2022)6:60

Response 2.

Thank you very much for your suggestion and the recommendation of insightful references. We added the four papers that the Reviewer 1 recommended to our reference list (we already had one paper among these four), and numbered them to be [21], [22], [23], and [24], as follows. (We omitted the reference number [79] which is renumbered to the reference number [22], and renumbered the rest of the remaining references.)

  1. M. Ye et al., "Spin coherence in two-dimensional materials", npj Computational Materials, 5, 44 (2019)
  2. A. Gottscholl et al., "Initialization and read-out of intrinsic spin defects in a van der Waals crystal at room temperature", Nature Materials, 19, 540–545 (2020)
  3. A. Gottscholl et al., "Room temperature coherent control of spin defects in hexagonal boron nitride", Science Advances 7, 14 (2021)
  4. J. Lee et al., ", First-principles theory of extending the spin qubit coherence time in hexagonal boron nitride", npj 2D Materials and Applications, 6, 60 (2022)

The spin coherence time of hBN color center compared to other material based color center was inserted in the Chapter 2. to compare NV center diamond and hBN to show the spinful nature of boron vacancy hBN color center. The inserted paragraphs are followings.

Page 6:

“The coherence times of the color centers in hBN are generally shorter than those of NV centers diamond. Unlike carbon atoms in diamonds, nitrogen and boron atoms in hBN have nuclear spins, which hinder the spin from being in coherent states. On the contrary, NV centers in diamond are renowned for their long coherence times at room temperature. In Ref [21] M. Ye et al. performed computer simulations to obtain the spin coherence times of four different 2D materials, namely delta-doped diamond layers, thin Si films, MoS2, and hBN. Compared to three other materials whose spin coherence times are around few miliseconds, hBN exhibited significantly short spin coherence times which are only about 10 ~ 30 microseconds, 2 order of magnitude smaller than the others. The boron-vacancy color center in hBN (which will be discussed again in Section 5-2) is well known for its spin texture, but the spin coherence lifetime is limited compared to that of NV center diamonds. The experimental study [22] tells us that the spin coherence lifetime of an hBN defect measured via Rabi oscillation was 10 microseconds at temperature T = 8K, while an NV-center diamond recorded a much longer spin coherence time of 400 microseconds even at room temperature.

             However, improvements in coherence times have been shown by careful engineering of the local environment and isotopically purifying the hBN samples. The study [23] demonstrated the coherent manipulation of VB− spinful color centers in hBN was possible even at room temperature, by applying pulsed spin resonance protocols. Moreover, at cryogenic temperature, spin-lattice relaxation time achieved the record of 18 microseconds, which is three orders of magnitude larger than its usual value. In Ref [24], the authors performed computation on the temporal properties of decoherence, by combining density functional theory (DFT) and cluster correlation expansion (CCE) and demonstrate that the coherence time can be extended by the factor of three, by replacing all the boron atoms in the hBN crystal to 10B isotopes.”

Comment 3.

Luminous efficiency: Already, as shown in the Ref. 19, specific luminous efficiency magnitudes have been discussed in several papers. Since luminous efficiency is very important for the applications, it is necessary to discuss the efficiency of each color center as quantitative as possible.

Response 3.

Thank you for your suggestion. The luminosity of hBN color center compared to other material based color center was inserted in the Chapter 2. The inserted sentences are followings.

Page 6,

“The luminosity factor of the color centers for quantum photonic applications is usually measured by the number of photons emitted from optically saturated single-photon emitters. In a previous study, a nitrogen-vacancy single-photon emitter [25] achieved 4.2 Mcps (million photon counts per second), showing compatibility with other materials such as NV-center diamond and SiC with a brightness of roughly 0.1~1 Mcps.

Comment 4.

Finally, the future prospects for the quantum application of the hBN color center should be discussed by comparing it with the color centers of other materials. This will attract the interest of many readers in the research of the hexagonal boron nitride color center.

Response 4.

As Reviewer 1 requested, at the Chapter 6. Conclusion we inserted paragraphs depicting future applications of hBN, empathizing the prominence of hBN compared to other materials such as NV center diamond and SiC in the field of quantum technologies and advanced UV optoelectronics. The inserted paragraphs are followings in page 15~16.

“Some of hBN’s key applications include, but are not limited to, quantum photonics and UV optoelectronics. In quantum photonics, hBN is being explored as a platform for on-chip integrated quantum photonic devices. It can be used to create sources of single photons from color centers, which are crucial for quantum information processing and quantum key distribution. Compared to other materials such as NV center diamond and silicon carbide (SiC), hBN has a great advantage in that it is an atomically thin 2D material; therefore, its integration into quantum photonic chips and the manipulation of optical properties, such as straining the device, are much easier.

In UV optoelectronics, hBN possesses a wide bandgap, making it an excellent candi-date for UV optoelectronic applications. It can be used to create efficient UV light emitters, detectors, and sensors. hBN-based LEDs can be used in advanced UV lighting applications, such as sterilization, water purification, and UV curing processes in industries. These applications highlight the broad potential of hBN in advancing quantum technologies and UV optoelectronics, enabling the development of more efficient, compact, and robust devices for various scientific and industrial applications.”

Comment 5.

Minor correction: Page 1 line36 graphite -> graphene

Since the crystal structure of a single-layer hexagonal boron nitride is described here, graphene (monolayer) should be used instead of graphite (multi layers).

Response 5.

Thank you for your suggestion. We replaced the word “graphite” to “graphene” as you requested.

Reviewer 2 Report

The manuscript “Color Centers in Hexagonal Boron Nitride”

This is not recommended in Nanomaterial MDPI journal due to following concerns regarding review article

1.     Page 1 line No. 33 “which” should be “which”

2.     Page 1 line No. 43 “allows the intrinsic charge carrier scattering limited high performance” should be rewritten as “allows for limited intrinsic charge carrier scattering”

3.     Maintain singular-plural throughout the article as in line No. 44 “material” should be “materials”

4.     The quality as well as size of the graphics included in the article need to be polished as page 2 Figure 1(d).

5.     Page No. 2 line No. 68 “shifts” should be replaced by “shifting”.

6.     Page 3 line No. 83 “due to their” should be corrected as “due to its”.

7.     There is much preposition error throughout the article author must revise the article grammatically like line 87 “over” should be “at”.

8.     Page 3 line No. 93 “knowns as” should be “known as”.

9.     Maintain the singular plural agreement throughout the article.

10.  Paraphrase lines 113-122 to make them fluent and more effective to understand. Consider them to make simpler ones.

11.  There is great noise in the peaks of photoluminescence which should be removed as shown in figure 8 (a).

12.  Quality of the SEM is very poor.

13.  In Figure 9 graphs interpretation needs to be rechecked and reconsidered as they need to be polished and rearranged.

14.  Summary of the article need to be rewritten it does not do just with the article it should be more precise and selective.

Moderate editing of English language required

Author Response

Comment from Reviewer 2. and the Author’s Response.

Comment 0.

This is not recommended in Nanomaterial MDPI journal due to following concerns regarding review article.

Response 0.

Thank you very much for pointing the crucial point to polish the paper. As the Reviewer 2 requested we applied correction to every single grammatical error point. Not only that, we applied English style correction using professional proof-reader provided by MDPI. Thank you for your suggestion.

Comment 1.

Page 1 line No. 33 “which” should be “which”

Response 1.

Recommended correction applied.

Comment 2.

Page 1 line No. 43 “allows the intrinsic charge carrier scattering limited high performance” should be rewritten as “allows for limited intrinsic charge carrier scattering”

Response 2.

Recommended correction applied.

Comment 3.

Maintain singular-plural throughout the article as in line No. 44 “material” should be “materials”

Response 3.

Recommended correction applied.

Comment 4.

The quality as well as size of the graphics included in the article need to be polished as page 2 Figure 1(d).

Response 4.

Recommended correction applied. We arranged the figures to provide better visibility, especially for the graph figures.

Comment 5.

Page No. 2 line No. 68 “shifts” should be replaced by “shifting”.

Response 5.

Recommended correction applied.

Comment 6.

Page 3, line No. 83 “due to their” should be corrected as “due to its”.

Response 6.

Recommended correction applied.

Comment 7.

There is much preposition error throughout the article author must revise the article grammatically like line 87 “over” should be “at”.

Comment 7.

Thank you for your suggestion. we applied correction to every single grammatical error point.

Comment 8.

Page 3, line No. 93 “knowns as” should be “known as”.

Response 8.

Recommended correction applied.

Comment 9.

Maintain the singular plural agreement throughout the article.

Response 9.

Recommended correction applied.

Comment 10.

Paraphrase lines 113-122 to make them fluent and more effective to understand. Consider them to make simpler ones.

Response 10.

Thank you for your suggestion. I hope the Reviewer 2 is pleased with the replaced paragraph in page 3 which is written as follows.

“The Franck–Condon principle explains the optical transition between the ground state and excited state of color centers. According to the Franck–Condon principle, the nuclei are considered fixed due to their much larger mass compared to the electrons during an electronic transition. However, the degree of freedom of nuclear motion can lead to changes in the vibrational energy levels taking into account the coupling between the electronic and vibrational modes. This explains the interplay between electronic and vibrational transitions in color centers in ultrawide-bandgap materials during processes such as the absorption, emission, or scattering of light, as shown in Fig. 2(d)

Comment 11.

There is great noise in the peaks of photoluminescence which should be removed as shown in figure 8 (a).

Response 11.

We also agree with the high noise level of the PL spectrum from oxygen atom based hBN color center. However, in order to avoid the noise, we have no choice but to delete the whole graph, and we decided to maintain the graph for it is essential data to explain the property of the oxygen atom based hBN color center.

Comment 12. 

Quality of the SEM is very poor.

Response 12.

I am sorry for any inconvenience. We agree with your assessment. We decided to omit the poor-quality SEM image in Fig 9 (renumbered to Fig 10)

Comment 13.

In Figure 9 graphs interpretation needs to be rechecked and reconsidered as they need to be polished and rearranged.

Response 13.

Thank you very much for pointing this out. We omitted two figures in Figure 9(d), rearrange for better visibility, and renamed as Figure 10 with the insertion of Figure 3, the NV center diamond. Here we show how the Figure 9 has been changed.

Before Revision

After Revision

Comment 14.  

Summary of the article need to be rewritten it does not do just with the article it should be more precise and selective.

Response 14.

Thank you for your suggestion. As Reviewer 2 requested, we inserted paragraphs depicting future applications of hBN in quantum technologies and advanced UV optoelectronics at the Chapter 6. Conclusion. The inserted paragraphs are followings in page 15~16.

“Some of hBN’s key applications include, but are not limited to, quantum photonics and UV optoelectronics. In quantum photonics, hBN is being explored as a platform for on-chip integrated quantum photonic devices. It can be used to create sources of single photons from color centers, which are crucial for quantum information processing and quantum key distribution. Compared to other materials such as NV center diamond and silicon carbide (SiC), hBN has a great advantage in that it is an atomically thin 2D material; therefore, its integration into quantum photonic chips and the manipulation of optical properties, such as straining the device, are much easier.

In UV optoelectronics, hBN possesses a wide bandgap, making it an excellent candi-date for UV optoelectronic applications. It can be used to create efficient UV light emitters, detectors, and sensors. hBN-based LEDs can be used in advanced UV lighting applications, such as sterilization, water purification, and UV curing processes in industries. These applications highlight the broad potential of hBN in advancing quantum technologies and UV optoelectronics, enabling the development of more efficient, compact, and robust devices for various scientific and industrial applications.”

Comment 15.

Comments on the Quality of English Language

Moderate editing of English language required

Response 15.

Thank you very much for your comment. As the Reviewer 2. requested, we applied English style correction using professional proof-reader provided by MDPI.

Reviewer 3 Report

The article "Color Centers in Hexagonal Boron Nitride" presents a review on color centers in hBN, where various aspects of hBN color centers, such as fundamental photon emission principles, categorization based on emitted wavelengths, fabrication methods, etc are considered. hBN color centers associated with defect are also reviewed. Some questions are raising when reading a review. They need to be answered before the work will be suggested for publication.

It will be good to see more information on the history of hBN discovery. When? How? What kind of technics were used? What substrates were used? etc.

Figure 1d, Figures 2a,d,f, Figure 9d are of a very bad quality. It is not possible for a reader to see the details. is of a very bad quality. It is not possible for a reader to see the details.

The title of the article is "Color Centers in Hexagonal Boron Nitride". However, part #2 of the article starts as follows "In this chapter, we will look over the behaviors of defects and impurities in ultra-wide bandgap semiconductors such as diamond and hBN.". Why is diamond considered? Why was it not mentioned in the part #1 of the article?

There is one big issue that may confuse the reader. It is not clear if the article is about 2D hBN, hBN crystal of hBN thin films. In different parts of a review there is a different information and no systematic presentation.

At the part #1 of the article the role of first-principles calculations in the estimation of the band structure of hBN is mentioned. However, no review on the role of first-principles calculations in study of the defects, for example, vacancy, substitutional, interstitial, and self-interstitial defects in hBN. In my opinion it is important to show significant impact of modelling to the investigation of hBN or make the review totally experiment oriented.

Can Authors elaborate how histogram of the ZPL wavelength of color centers in hBN (Fig 4.) has been created? Are only experimental data collected there? Why are 148 data points selected?

The Conclusions seems to be very short. It is suggested to enlarge the discussion with the potential applications of hBN in quantum technologies and advanced UV optoelectronics.

Author Response

"To see the figures, Please see the attachment." 

Comment from Reviewer 3. and and the Author’s Response.

Comment 0.

The article "Color Centers in Hexagonal Boron Nitride" presents a review on color centers in hBN, where various aspects of hBN color centers, such as fundamental photon emission principles, categorization based on emitted wavelengths, fabrication methods, etc are considered. hBN color centers associated with defect are also reviewed. Some questions are raising when reading a review. They need to be Responseed before the work will be suggested for publication.

Response 0.

Thank you very much for your discussion and raising questions on our review paper. We totally agree with Reviewer 3’s opinion to polish our explanation about hBN color centers and other materials.

Comment 1.

It will be good to see more information on the history of hBN discovery. When? How? What kind of technics were used? What substrates were used? etc.

Response 1.

Thank you for your suggestion. As the Reviewer 3 requested, we inserted the advance of HTHP (High-Temperature High-Pressure) hexagonal boron nitride synthesis technique invented by Takashi Taniguchi and his colleagues, and the spectroscopic confirmation using CL (Cathodoluminescence) spectroscopy.

The modified paragraphs are followings in page 2~3.

“To create high-quality hBN quantum devices, it is essential to use hBN with excellent material quality. Efforts to grow hBN have been attempted extensively, but challenges re-lated to crystallinity issues and high impurity contents have often led to failures. In 2004, a group of scientists in NIMS (National Institute for Materials Science, Japan) led by T. Taniguchi [8] successfully achieved the large-scale growth of hBN using the HTHP (high-temperature high-pressure) method. The authors used the temperature gradient method under HP (4.0-5.5GPa)/HT (1500-1700°C) conditions using barium boron nitride (Ba3B2N4) as a solvent system to prepare samples of deoxidized hBN. They confirmed the high crystallinity and low defect density with efficient deep UV emission and lasing be-havior at a photon energy of 5.7 eV (215 nm), as shown in Fig. 1(d), and claimed that mul-tilayer hBN has a direct bandgap. The ultraviolet emission spectrum of multilayer hBN was acquired using cathodoluminescence (CL) spectroscopy, irradiating an electron beam on the sample to excite the valence band electrons and observing the light emission during the electron–hole recombination. Despite the efficient and intense emission from multi-layer hBN around 5.7 eV (215 nm), the basic question of the nature of the bandgap proper-ties and bandgap value of hBN was controversial. In 2016, G. Cassabois et al. presented evidence that hBN has an indirect bandgap, along with evidence of a phonon-assisted op-tical transition at 5.955 eV with 130 meV exciton binding energy, through two-photon spectroscopy and temperature-dependent photoluminescence [9]. Nevertheless, hBN has emerged as a key material for the development of robust, next-generation optoelectronics due to its large bandgap (close to 6 eV, and even larger in some cases) and efficient pho-non-assisted optical transition [10] ~ [13].”

Comment 2.

Figure 1 (d), Figures 2 (a), (d), (f), Figure 9 (d) are of a very bad quality. It is not possible for a reader to see the details. is of a very bad quality. It is not possible for a reader to see the details.

Response 2.

I am sorry for any inconvenience. As the Reviewer 3 requested, we arranged the figures for better visibility. In order to enlarge Figure 1 (d), we reduced the size of Figure 1 (a), (b). We revised the Figure 2 (e), (f) and rearranged the Figure 2. We omitted two figures in Figure 9 (d), rearranged, and renamed as Figure 10. with the insertion of Figure 3, the NV center diamond. Here we show how the Figure 1, 2, and 9 has been changed.

Fig 1.

Before Revision

After Revision

Fig 2.

Before Revision

After Revision

Fig 9.

Before Revision

After Revision

Comment 3.

The title of the article is "Color Centers in Hexagonal Boron Nitride". However, part #2 of the article starts as follows" In this chapter, we will look over the behaviors of defects and impurities in ultra-wide bandgap semiconductors such as diamond and hBN.". Why is diamond considered? Why was it not mentioned in the part #1 of the article?

Response 3.

Thank you for providing these insights. In Chapter 2. “Color centers in ultra-wide bandgap semiconductor”, in order to investigate the hBN color center, we first wanted to explain the theory of color centers in general semiconductor with wide bandgap, not only limited to hBN. We considered diamond color center to be our prerequisite for studying hBN color center, since the researches on NV center Diamond and quantum photon source based on it will help us understanding hBN color centers. In the revised version of the manuscript, we inserted some paragraphs explaining the properties of diamond and comparison with hBN. The inserted paragraphs are followings, in page 5~6.

“In order to provide specific information, we compare nitrogen-vacancy centers in diamond with the color centers in hexagonal boron nitride. The research on quantum de-vices using nitrogen-vacancy (NV) centers in diamond (Fig 3. (a)) [17] dates back to the late 1990s when NV centers were first observed and characterized. The NV center in diamond is a point defect composed of a substitutional nitrogen atom adjacent to a vacant lattice site (vacancy). The detection of electron paramagnetic resonance (EPR) from a single NV center led to the study of quantum devices. Researchers discovered that NV centers possess unique properties, such as long electron spin coherence times even at room temperature, making them promising candidates for quantum information processing and sensing applications. [18] [19]

Throughout the 2000s, significant progress was made in understanding and manipulating the quantum properties of NV center diamond, including quantum control of the electron spin, initialization, and readout techniques. Despite the promising features, NV-center diamond also has limitations. One of the main challenges is related to the coherence times, which can be affected by the surrounding environment and impurities in the crystal lattice. The scalability of NV-center-diamond-based quantum devices has been hindered due to difficulties in fabricating large-scale devices and integrating them with other quantum components. In recent years, researchers have turned their attention to hBN as an alternative platform for quantum devices. hBN's two-dimensional nature and unique crystal structure make it an attractive candidate for quantum technologies, especially for integration with quantum photonic circuits. As a result, efforts have been made to study color centers in hBN as a potential replacement for NV centers in diamond.

The band structure of diamond (Fig 3. (b)) [18] [19] is characterized by a large bandgap be-tween the valence band and the conduction band. The NV center diamond introduces lo-calized energy levels within the bandgap, creating an energy level scheme that involves electronic transitions between these levels. The NV center diamond has a ground state and two low-lying excited states, separated by a zero-phonon line (ZPL). The ZPL wavelength of the NV center in diamond is around 637 nm (nanometers) in the visible region of the electromagnetic spectrum. The ground state corresponds to the electronic configuration of a nitrogen atom in a substitutional site in a diamond lattice with an unpaired electron spin. The two excited states are associated with transitions involving the nitrogen spin and lattice vibrations (phonons).

The NV center diamond's energy level structure exhibits spin-dependent optical transitions, meaning that its optical properties are sensitive to the spin state of the electron. This spin-dependent nature allows for an efficient and high-fidelity readout of the NV center's electron spin state, a crucial property for quantum information processing. In conclusion, the NV center in diamond possesses unique properties related to its electronic bandgap, such as spin-dependent optical transitions and a sharp zero-phonon line emission peak. [20] Comparing NV centers in diamond and color centers in hexagonal boron nitride, hBN has a wider bandgap in the ultraviolet range, typically around 6 eV. The color centers in hBN exhibit broadband photoluminescence in the visible and UV regions. Diamond also has a large bandgap of approximately 5.5 eV, and the NV center diamond exhibits well-defined optical transitions with a zero-phonon line (ZPL) at around 637 nm in the visible spectrum.

The coherence times of the color centers in hBN are generally shorter than those of NV centers diamond. Unlike carbon atoms in diamonds, nitrogen and boron atoms in hBN have nuclear spins, which hinder the spin from being in coherent states. On the contrary, NV centers in diamond are renowned for their long coherence times at room temperature. In Ref [21] M. Ye et al. performed computer simulations to obtain the spin coherence times of four different 2D materials, namely delta-doped diamond layers, thin Si films, MoS¬2, and hBN. Compared to three other materials whose spin coherence times are around few miliseconds, hBN exhibited significantly short spin coherence times which are only about 10~30 microseconds, 2 order of magnitude smaller than the others. The boron-vacancy color center in hBN (which will be discussed again in Section 5-2) is well known for its spin texture, but the spin coherence lifetime is limited compared to that of NV center diamonds. The experimental study [22] tells us that the spin coherence lifetime of an hBN defect measured via Rabi oscillation was 10 microseconds at temperature T = 8K, while an NV-center diamond recorded a much longer spin coherence time of 400 microseconds even at room temperature.

             However, improvements in coherence times have been shown by careful engineering of the local environment and isotopically purifying the hBN samples. The study [23] demonstrated the coherent manipulation of VB− spinful color centers in hBN was possible even at room temperature, by applying pulsed spin resonance protocols. Moreover, at cryogenic temperature, spin-lattice relaxation time achieved the record of 18 microseconds, which is three orders of magnitude larger than its usual value. In Ref [24], the authors performed computation on the temporal properties of decoherence, by combining density functional theory (DFT) and cluster correlation expansion (CCE) and demonstrate that the coherence time can be extended by the factor of three, by replacing all the boron atoms in the hBN crystal to 10B isotopes.

The luminosity factor of the color centers for quantum photonic applications is usually measured by the number of photons emitted from optically saturated single-photon emitters. In a previous study, a nitrogen-vacancy single-photon emitter [25] achieved 4.2 Mcps (million photon counts per second), showing compatibility with other materials such as NV-center diamond and SiC with a brightness of roughly 0.1 ~ 1 Mcps.

Comment 4.

There is one big issue that may confuse the reader. It is not clear if the article is about 2D hBN, hBN crystal of hBN thin films. In different parts of a review there is a different information and no systematic presentation.

Response 4.

I am sorry for giving any confusion to the Reviewer. In this paper we are reviewing many different experimental results, some of them use 2D hBN usually grown by CVD method, other uses mechanically exfoliated hBN flakes from hBN crystal, synthesized by HPHT (High-Temperature High-Pressure) method.

Comment 5.

At the part #1 of the article the role of first-principles calculations in the estimation of the band structure of hBN is mentioned. However, no review on the role of first-principles calculations in study of the defects, for example, vacancy, substitutional, interstitial, and self-interstitial defects in hBN. In my opinion it is important to show significant impact of modelling to the investigation of hBN or make the review totally experiment oriented.

Response 5.

We totally understand what the Reviewer 3 concerned about, and we agree that it would be the best case to show the DFT calculation result of every single kind of the hBN color centers. However, we decided to show up only the DFT calculation result of pristine hBN and make the rest of the review on hBN color centers largely experiment oriented, since the exact origin of hBN color center emissions are still unexplored so the detailed calculation works to forecast the energy levels are still undergoing. The only kind of the hBN color center under detailed computational investigation is carbon based one, where simulated PL emission peaks are available for different carbon atom numbers 2, 4, and 6, as shown in Fig 10 (b).

Comment 6.

Can Authors elaborate how histogram of the ZPL wavelength of color centers in hBN (Fig 4.) has been created? Are only experimental data collected there? Why are 148 data points selected?

Response 6.

The histogram of the ZPL wavelength has been created by only selecting experimentally observed spectroscopy data from the reference [6] ~ [94]. The 148 data points are selected from spectrum graphs that the authors of the papers identified the wavelength. The number of data point per paper might vary.

Comment 7.

The Conclusions seems to be very short. It is suggested to enlarge the discussion with the potential applications of hBN in quantum technologies and advanced UV optoelectronics.

Response 7.

As Reviewer 3 requested, we inserted paragraphs depicting future applications of hBN in quantum technologies and advanced UV optoelectronics at the Chapter 6. Conclusion, as follows.

“Some of hBN’s key applications include, but are not limited to, quantum photonics and UV optoelectronics. In quantum photonics, hBN is being explored as a platform for on-chip integrated quantum photonic devices. It can be used to create sources of single photons from color centers, which are crucial for quantum information processing and quantum key distribution. Compared to other materials such as NV center diamond and silicon carbide (SiC), hBN has a great advantage in that it is an atomically thin 2D material; therefore, its integration into quantum photonic chips and the manipulation of optical properties, such as straining the device, are much easier.

In UV optoelectronics, hBN possesses a wide bandgap, making it an excellent candi-date for UV optoelectronic applications. It can be used to create efficient UV light emitters, detectors, and sensors. hBN-based LEDs can be used in advanced UV lighting applications, such as sterilization, water purification, and UV curing processes in industries. These applications highlight the broad potential of hBN in advancing quantum technologies and UV optoelectronics, enabling the development of more efficient, compact, and robust devices for various scientific and industrial applications.”

Reviewer 4 Report

The manuscript is intended as a concise review on the photoluminescence of defective hBN samples. In particular, variants of color centers artificially produced in hBN are considered and their spectral properties summarized along with the most popular fabrication methods.

The topic of luminescence in wide band gap 2D materials due to color centers is sound, mostly because of the perspective use of such samples in single photon quantum emitters.

While the mechanisms behind the achievement of specific spectral features and their correlation with the color center structure are not discussed in details, nor the polarization properties addressing exploitations in quantum technologies are reported in detail, the paper is a useful and easy to read reference for the community involved in the field.

The manuscript is clear and, despite its conciseness, can be a viable summary of the present state-of-the-art in hBN color centers.

Therefore, my opinion is that the paper can deserve publication.

Prior to acceptance, Authors are requested to account for the following minor points.

1.     Language and style must be improved. There are many instances of wrong/incomplete sentences. Authors are strongly recommended to revise the text with the help of a professional proof-reader.

2.     Readability of figures is cumbersome. I understand this is because graphs are taken from already published papers, where more space was available for graphical contents. However, Authors may consider slightly modifying the layout and organize figures such that a smaller number of columns, and a larger number of rows, are used (I mean, rather than having three panels in a row and two rows, they can for instance modify in order to have two panels in a row and three rows).

3.     I am not convinced that “> 6 eV” at line 63 is correct. I think it should read “close to 6 eV, and even larger in some cases”.

4.     The occurrence of a broad PL peak in the case of single layer hBN containing N-vacancies, mentioned at line 305, would deserve some comment aimed at providing the readers with some physical interpretation.

5.     The D quantity at line 339 must be defined.  

English must be thoroughly revised for the presence of many incomplete or wrongly organised sentences.

Author Response

"To see the figures, please see the attachments."

Comment from Reviewer 4. and the Author’s Response.

Comment 0.

The manuscript is intended as a concise review on the photoluminescence of defective hBN samples. In particular, variants of color centers artificially produced in hBN are considered and their spectral properties summarized along with the most popular fabrication methods.

The topic of luminescence in wide band gap 2D materials due to color centers is sound, mostly because of the perspective use of such samples in single photon quantum emitters.

While the mechanisms behind the achievement of specific spectral features and their correlation with the color center structure are not discussed in details, nor the polarization properties addressing exploitations in quantum technologies are reported in detail, the paper is a useful and easy to read reference for the community involved in the field.

The manuscript is clear and, despite its conciseness, can be a viable summary of the present state-of-the-art in hBN color centers.

Therefore, my opinion is that the paper can deserve publication.

Prior to acceptance, Authors are requested to account for the following minor points.

Response 0.

Thank you very much for your fruitful discussion on our review paper. We deeply appreciate your comment and concern, and we totally agreed with Reviewer 4’s opinion and reflected any requests to the revised paper.

Comment 1.

Language and style must be improved. There are many instances of wrong/incomplete sentences. Authors are strongly recommended to revise the text with the help of a professional proof-reader.

Response 1.

We agree with you. As the Reviewer 4. requested, we applied English style correction using professional proof-reader provided by MDPI.

Comment 2.

Readability of figures is cumbersome. I understand this is because graphs are taken from already published papers, where more space was available for graphical contents. However, Authors may consider slightly modifying the layout and organize figures such that a smaller number of columns, and a larger number of rows, are used (I mean, rather than having three panels in a row and two rows, they can for instance modify in order to have two panels in a row and three rows).

Response 2.

We agree with you. As the Reviewer 4. requested, we rearranged the figures to reduce the number of figures per row. Furthermore, enlarged the graphs and reduced the size of unnecessarily large (i. e. atomic schematics) figures to avoid any confusion reading the figure. Here is an example, showing how Figure 9 has been changed for better visibility.

Fig 9.

Before Revision

After Revision

Comment 3.

I am not convinced that “> 6 eV” at line 63 is correct. I think it should read “close to 6 eV, and even larger in some cases”.

Response 3.

Thank you for providing these insights. The bandgap >6eV comment was changed as the Reviewer 4 requested. We inserted the sentence “close to 6 eV, and even larger in some cases” to the abstract.

Comment 4.

The occurrence of a broad PL peak in the case of single layer hBN containing N-vacancies, mentioned at line 305, would deserve some comment aimed at providing the readers with some physical interpretation.

Response 4.

Thank you for point the crucial fact. The hBN sample at the reference [19] was prepared from CVD-grown hBN on the Cu-foil, and wet-transferred to the Silicon substrate. We inserted the following physical interpretation to the revised manuscript at page 10 as follows.

“We attribute the broader phonon sideband emission from the monolayer hBN to the stronger phonon interaction with the substrate in monolayer hBN crystal structure, where the atomically thin monolayer hBN is more vulnerable to electron-phonon interaction.”

Comment 5.

The D quantity at line 339 must be defined.

Response 5.

We agree with your opinion. The zero-field splitting value of D was calculated and explained in page 12 as follow.

The important factor describing the negatively charged boron-vacancy color center in hBN, VB-, is the zero-field splitting (ZFS) splitting D value, which is the energy splitting between the state mS = 0 and the states mS = ± 1 (degenerated, same values). The value of ZFS is D = 14 μeV, which corresponds to D/h ~ 3.5 GHz.”

Round 2

Reviewer 2 Report

Authors have updated their manuscript according to the suggestions so it maybe accepted in its current form.

Minor editing of English language required

Reviewer 3 Report

Thanks to the Authors for replying the comments. All issues seems to be resolved. The review is suggested for publication.